# My Parents Taught…Green Was My Growth! The Role of Intergenerational Transmission of Ecological Values in Young Adults’ Pro-Environmental Behaviors and Their Psychosocial Mechanisms

**DOI:** 10.3390/ijerph19031670

**Published:** 2022-02-01

**Authors:** Massimiliano Scopelliti, Daniela Barni, Elena Rinallo

**Affiliations:** 1Department of Human Studies, Libera Università Maria Ss. Assunta (LUMSA University), 00193 Rome, Italy; e.rinallo@lumsa.it; 2Department of Human and Social Sciences, Università degli Studi di Bergamo, 24129 Bergamo, Italy; daniela.barni@unibg.it

**Keywords:** pro-environmental behavior, ecological values, affect, identity, intergenerational transmission, fathers, mothers

## Abstract

Past research on pro-environmental behaviors (PEBs) has identified several psychosocial determinants, ranging from personal values to attitudes—mostly environmental concerns—and norms. Less attention has been devoted to the role of affect and identity processes, until recently, when investigations began into the emotional connections with nature and environmental identity, i.e., one’s self-concept in relation to the natural world. Finally, research into the parent–child transmission of ecological values was recently developed. We aimed to analyze the role of the above-mentioned variables in predicting different PEBs, within a comprehensive framework. We hypothesized a chain relationship between the ecological values of parents and mothers, the ecological values of their children, environmental concerns, affect towards nature, environmental identity, and PEBs, as the final outcomes. In a cross-sectional exploratory study, an online questionnaire was administered to 175 young Italian adults. Validated scales to measure the above variables and socio-demographics were included. The results showed a different pattern of predictors for each PEB. Overall, the importance of the emotional connection with nature and environmental identity in predicting PEBs has clearly emerged. Finally, the role of intergenerational transmission of ecological values in PEBs, with differences between the influence of fathers and mothers, is outlined. The study provides a more integrative view of PEBs by considering the variety of human processes. Theoretical and practical implications of results are discussed.

## 1. Introduction

Pro-environmental behavior (PEB) has long been considered a moral issue [1,2] and it has been referred to in many different ways in psychological literature, including “environmentally-concerned behavior” [3], “environmentally significant behavior” [4], “environmentally responsible behavior” [5], “ecological behavior” [6,7], and “sustainable behavior” [8,9]. Overall, there is wide agreement across the above terms that PEB implies friendly actions towards the environment, resulting in the protection of its well-being. From a psychological perspective, Stern [4] convincingly discussed the difference between the impact-oriented and the intent-oriented approach in understanding PEB. The first has to do with the consequences of human action on the environment (e.g., car use and environmental pollution); the second implies the role of human intentions behind the behavior, namely what impacts people expect to have on the environment through willing actions (e.g., avoiding buying plastic products). The author has stressed that “environmental intent may fail to result in environmental impact” (p. 408); it is thus essential to identify target behaviors, whose impacts on the environment can be huge, and then adopt an intent-oriented approach to understand the psychological determinants of those conducts.

Among the most investigated PEBs that have shown strong positive impacts on the environment, we will emphasize the roles of environmental activism [10], resource conservation, with energy and water conservation receiving particular attention [11,12,13,14,15,16], waste reduction and recycling [17,18,19], sustainable mobility [20,21], and sustainable consumption [22,23,24,25]. Overall, the role of environmental well-being for human well-being has been largely recognized as well [26,27,28].

Another key issue has to do with the measurement of PEB. In this regard, Lange and Dewitte [29] recently debated the existing measures, how they have been applied, their strengths and weaknesses, and potential for improvement. A relevant distinction was made between behavioral tasks to be used in an experimental setting, self-reports, and observations in ecological conditions. Beyond the role of biases and ecological validity, the choice between them depends on several aspects, including the broader vs. narrower focus of PEB under investigation. Although the prediction of actual behavior through self-reports should take into account a variety of relevant moderators [30], the authors have pointed out that when different PEBs and their psychological determinants have to be investigated, the use of self-reports can be highly recommended. Going further in this reflection, Lange and Dewitte [29] stressed the importance of using multi-item validated scales vs. single-item measures to better “contributing to a cumulative science of PEB” (p. 93). Among the multi-item scales developed over time, the authors identified 20 validated tools for the measurement of PEB in general, encompassing different dimensions—or domains—of PEB. With reference to the adult population, Karp [31] developed the environmental behavior scale (16 items), investigating activism, good citizenship, and healthy consumption; Stern, Diez, Abel, Guagnano, and Kalof [32] used the environmentalism scale (16 items), addressing committed activism and three dimensions of support for the environmental movement, namely consumer behavior, willingness to sacrifice, and environmental citizenship; through several studies, Kaiser and colleagues [33,34,35] developed, validated, and refined the general ecological behavior scale (40 items in the shorter version), measuring energy and water conservation, mobility and transportation, waste avoidance, recycling, consumerism, and vicarious behaviors toward conservation. In a cross-cultural research, Schultz et al. [36] administered the environmental behavior scale (12 items), referring to a variety of different PEBs included in a single dimension. Markle [37] developed the pro-environmental behavior scale (19 items), with four dimensions, labelled conservation, environmental citizenship, food, and transportation. In a quali–quantitative study among landowners, hunters, and birdwatchers, Larson, Stedman, Cooper, and Decker [38] validated the pro-environmental behavior scale (13 items), with the four dimensions of conservation lifestyle behaviors, land stewardship behaviors, social environmentalism, and environmental citizenship behaviors. Lange and Dewitte [29] argued that “based on its frequency of use and thoroughness of psychometric evaluation, the General Ecological Behavior (GEB) measure can probably be considered the best established of these domain-general propensity measures” (p. 94).

### 1.1. Values and Other Psychosocial Determinants of Pro-Environmental Behavior

The roles of human behavior in environmental problems, such as pollution [39,40,41,42,43], waste of resources [44], loss of biodiversity [45,46,47], and overall climate change [48], have been extensively documented. Thus, it is fundamental to understand the psychological determinants of those PEBs whose impacts on the above environmental problems can be more pronounced [49]. In this regard, Steg and Vlek [50] proposed an integrative framework on pro-environmental motivation and behavior, suggesting a link between the identification of PEBs that significantly affect environmental quality, relevant psychosocial determinants, interventions, and evaluation of efficacy. The authors have distinguished between three motivational factors towards PEBs, namely moral and normative concerns, costs and benefits, and affect.

Both moral and normative concerns and cost/benefit weighting have to do with cognitions; that is, what people think about PEBs, the evaluated importance and consequences in personal or social terms. Within this domain of psychosocial determinants of PEBs, the literature has identified several relevant constructs. Personal values have been defined as the beliefs on desirable and trans-situational goals, with different importance, acting as guiding principles in people’s lives [51]. The universal structure of human values has been consistently shown [52,53,54] and, according to the value attitude behavior model [55], values are at the base of a cognitive hierarchical mechanism leading to behavior that influences attitudes and norms through a ‘value orientation’. Specifically, the role of personal values in PEBs has been found in a variety of empirical studies. For example, Schultz and Zelezny [56] showed that personal values referring to self-transcendence are important predictors of PEBs across five countries. In Brown and Kasser’s study [57], a positive relationship between altruistic values and PEBs emerged. Nordlund and Garvill [58] found a relationship among self-transcendence, environmental values, personal norms, and PEBs. In a cross-national study across 27 countries, Oreg and Katz-Gerro [59] outlined an indirect relationship between post-materialistic values [60] and PEBs. Kaiser, Ranney, Hartig, and Bowler [61], with specific reference to values related to the environment, predicted pro-environmental intentions, which in turn predicted PEBs in two studies involving different social groups in the Swiss and in the US. deGroot and Steg [62] found empirical evidence of a relevant distinction among egoistic, altruistic, and biospheric values for understanding environmental beliefs, intentions, and PEBs, with biospheric values explaining best the pro-environmental cognitive and behavioral dependent variables. deGroot and Steg [63] also argued how altruistic and biospheric values can be considered as more stable predictors of PEBs, but also egoistic values can be compatible with pro-environmental issues, especially when the perceived conflict between them is reduced. Moreover, Nguyen, Lobo and Greenland [64] found a positive effect of biospheric values on purchasing behavior of energy efficient appliances. The role of values has also been consistently supported with reference to more committed PEBs, such as environmental activism [65,66,67,68]. 

Recently, a few studies have extended the attention to family values in relation to environmentally-related behaviors. From the national survey carried out by Oh et al. [69] involving 1519 respondents (18 years and above) in Singapore, it has emerged that biospheric (e.g., protection of the environment) and altruistic (e.g., helpful) family values—measured by asking “How important are the following principles in your family?”—were directly and positively related to a connection to nature and experiences of nature (i.e., the frequency and duration of urban greenspace and private or community garden use), while egoistic (e.g., social power) family values had a direct but negative association. The relationship between biospheric values and the connection to nature was also significantly mediated through social norms of family and friends, and personal experiences of nature.

Environmental concern has also been taken into account when explaining PEBs. Environmental concern has been conceived as an attitude towards facts and behaviors with negative consequences for the environment [70]. The most frequently used psychometric tool for the measurement of environmental concern is the new ecological paradigm scale (NEP) [71,72], referring to humans’ ability to upset the balance of nature, the existence of limits to growth, and humans’ right to rule over nature. More recently, Corral-Verdugo, Carrus, Bonnes, Moser, and Sinha [73] proposed a different tool, the new human interdependence paradigm scale (NHIP), suggesting and measuring an integrative vs. dichotomic vision of human needs and nature conservation. Environmental concern has emerged as a relevant construct in predicting PEBs in several studies [74,75,76,77]. However, the relationship between environmental concern and PEBs is usually not strong, because of the moderation or mediation effect of other variables. To give an example of the former, in a study involving respondents from 32 countries, Tam and Chan [78] found a role of cultural barriers in the association between environmental concern and PEBs, so that the size of this association varies across nations. Gifford and Nilsson [79] recently proposed a review on these variables. Norms have been frequently investigated as examples of the latter, and are included in comprehensive theories, such as the Norm Activation Model (NAM) [80], and the Value-Belief-Norm Theory (VBN) [32]. Within these frameworks, personal norms have emerged as the ultimate, and more direct determinants of PEBs in a causal chain, with consistent empirical evidence [63,81,82,83,84]. Subjective norms, referring to the expectations of significant others, have been considered as well in the Theory of Reasoned Action (TRA) [85] and its extension, the Theory of Planned Behavior (TPB) [86]. These theories best represent the costs/benefit analysis in psychological terms proposed by Steg and Vlek [50]. The predictive power of these theories on PEBs, also in association with VBN variables, has been repeatedly shown [7,59,87,88,89,90]. Moreover, the influence of norms on PEBs has been further investigated with reference to a distinction between several constructs, such as injunctive and descriptive norms [91,92,93,94], or local norms [95,96]. Farrow, Grolleau, and Ibanez [97] recently provided a review of different conceptualizations and empirical evidence on the effectiveness of social norms on PEBs.

Finally, Steg and Vlek [50] explored a role for affective processes, with the case of car use as the first PEB considered in empirical investigation. In this regard, Nilsson and Küller [98] showed a negative relationship between car affection and the acceptance of policy measures to reduce car use. Steg [99] found a stronger importance of symbolic and affective vs. instrumental motives for car use among commuters. In Carrus, Passafaro, and Bonnes’s study [100], a significant effect of anticipated negative emotions (e.g., feeling angry, frustrated, unsatisfied, discontented, guilty, sad) in reducing the use of private cars has also emerged. Moons and De Pelsmacker [101] reported on a significant role of emotions in the use of electric cars among Belgian drivers, even though peculiarities for different consumer segments.

Based on recent literature, this framework has possibilities for further developments. First, identity referring to a specific PEB has emerged as an added variable, improving the predictive power of different models, including TPB. This was shown in several studies analyzing recycling [102], ecological consumption [103,104], water or energy conservation [105,106,107], and activism [108]. More interestingly, the influence of a general environmental identity in PEBs has been proposed, referring to the aspect of one’s self-concept in relation to the natural world [109]. In this regard, Stern et al. [32] argued that “the step towards intense activism involves a substantial and transformational commitment, including a reframing of key elements of identity” (p. 84). More recently, several authors have stressed that PEBs in general are better predicted when outer variables of social influence have been internalized, as in the case of environmental identity [110,111,112]. As a consequence, the effect of norms can be reconsidered, taking into account the role of identity as a more proximal variable predicting PEBs. From this starting point, Whitmarsh and O’Neill [113] have shown a wider role of the pro-environmental identity on PEBs, referring to several everyday behaviors, such as waste reduction, eco-shopping and eating, water, and domestic energy conservation. Gatersleben et al. [103] found a significant effect of pro-environmental identity on recycling, buying fair trade, and avoiding flying on holidays. The importance of environmental identity on PEB has also been shown with reference to volunteering in stewardship activities in local parks [114]. Going further in the psychological mechanisms behind PEBs, a mediation role of pro-environmental self-identity has been recently outlined in the relationship between biospheric values and PEBs [115,116]. In a longitudinal study, Carfora, Caso, Sparks, and Conner [117] found a moderating effect of pro-environmental self-identity on several PEBs, including reducing food waste, food waste recycling, food packaging recycling, and food purchase.

Second, while affect towards specific PEBs can play a role, the role of a general affective relationship with the environment is presumably a relevant further variable to be considered in a comprehensive framework. In this regard, connectedness to nature and its psychometric measure (CNS) have been conceived as emotional connections with the natural world [118]. Similarly, the nature relatedness scale (NRS) [119] assesses the affective, cognitive, and experiential aspects of an individual’s connection to nature, and “encompasses one’s appreciation for and understanding of our interconnectedness with all other living things on the earth” (p. 718). Tam [120] proposed and empirically validated a measure of dispositional empathy towards nature (DEN), which refers to the dispositional tendency to understand and share the emotional experience of the natural world. The role of emotional bonds with nature on PEBs has emerged in previous research. For example, Hoot and Friedman [121] reported a significant effect of CNS on a six-item measure of PEBs, including signing petitions for or contributing to environmental causes, product selection based on environmental attributes, voting for political candidates for environmental reasons, membership in environmental groups, and reading publications by environmental groups. Martin et al. [122] found a positive relationship between connection to nature measured through NRS and PEBs, referring to both household and nature conservation. A similar association has emerged in other studies [123,124]. Geng, Xu, Ye, Zhou, and Zhou [125] considered a measure of both implicit and explicit connections through the Implicit Association Test [126] and the CNS, and they found a significant association of the former with spontaneous PEBs (i.e., the use of a plastic bag in the laboratory) and the latter with deliberative PEBs, measured through the GEB scale.

### 1.2. The Role of Parents and Intergenerational Transmission in Children’s Pro-Environmental Behavior

As already mentioned, PEBs are influenced by a “relevant other’s” expectations and actions [127]. Studies on intergenerational transmission of environmentalism for youngsters are consistent in assuming that parents are the primary sources of PEB development of their children and represent the main (and, for the first years of development, exclusive) social context of children’s pro-environmental practices [128]. Most of these studies have reported significant parent–child similarities in pro-environmental values, norms, concerns, attitudes, and behaviors [129,130,131], by showing that children who grew up with pro-environmental parents are likely to engage in more pro-environmental behavior as young adults [132].

Both direct and indirect intergenerational transmission processes in the environment domain are supported by research results. For example, Gong et al. [133] recently carried out a study on these processes involving a large sample of Chinese families with an early adolescent child (10–15 years). They pointed out that both mothers’ and fathers’ green consumption values were positively associated with adolescents’ green consumption values (i.e., direct transmission). This association was mediated by each parent’s environmentally responsible consumption behavior (i.e., indirect transmission), but only when there was a close parent–child relationship. In her research based on representative data from the parent–child socialization study in Belgium (2012), Meeusen [131] found that environmental attitudes were significantly related among father, mother, and adolescent child. On the other hand, the author showed that parent–child transmission of environmental concerns passed through the communication about the environment within the family. In those families that frequently talked about the environment, environmental concerns were more effectively transmitted from parents to adolescents. Interestingly, parent–child communication, and more in general parental behavior, had diverse influences depending on the type of pro-environmental behavior. From Matthies, Selge, and Klöckner’s study [94] on two relevant pro-environmental behaviors in Germany, it emerged that parents influenced the recycling behavior of their school-aged children (age 8–10) via sanctions and their own behavior, while re-use of paper was mainly influenced via communication of problem knowledge.

Research indicates that the parental influence on daily pro-environmental behavior of their children is universal, even though some differences between individualistic and collectivistic cultures have been identified [134,135]. Ando et al. [128], in their research with German and Japanese school-age children, found that environmental behavior of parents (e.g., paper recycling) directly affected the behavior of children, both in Germany and Japan. The findings also showed that parents’ behavior can influence children’s behavior by affecting the perception of seriousness of waste and subjective norms, especially in Japan (i.e., a collectivistic country), and personal norms, but only in Germany (i.e., an individualistic country). Subjective norms have been simply considered by the authors as the children’s experienced expectations of their parents; personal norms have been defined as the feeling of personal moral obligation based on the individual’s personal values, according to Schwartz’s Norm Activation model [80].

Personal values have been widely considered a central variable in the family relationships and intergenerational transmission processes [136,137], and, as mentioned above, their importance in PEBs has been outlined by several empirical studies [56,57,58]. Parent–child value similarity is the result of a complex network of mutual influences among parents, children, and their shared environments [138]. Transmission is widely acknowledged to be bidirectional with children having an active role and being able to influence their parents [139]. Children, indeed, perceive their parents’ values and can accept or reject what is perceived to be the parent’s viewpoint [140,141]. Perceived parental values have been found to contribute to predict children’s behaviors, such as academic and social behaviors [142], from childhood to young adulthood.

Fathers and mothers play specific (and non-interchangeable) roles in the transmission of values [143]. Mothers, who are generally more involved with the daily task of child-rearing and household, seem to be able to share more values with their children than fathers, so much that some studies have supported the hypothesis of a “female lineage” in the intergenerational transmission of values [140,144]. Mothers are usually carriers of relational values, such as benevolence and conservation, while fathers tend to give more importance to action values, such as power and stimulation [145]. Consistently, Power and Shanks’ qualitative study [146] found that mothers are likely to encourage interpersonal behaviors such as manners and politeness, and adolescents’ involvement in domestic chores, while fathers are more involved in encouraging instrumental behaviors such as assertiveness and independence.

Despite all of this, there has been little examination of how fathers’ and mothers’ ecological values influence children’s ecological values and pro-environmental behavior development [147]. Most studies have indeed focused on intergenerational transmission of general values rather than of environmental core values [130].

### 1.3. Overview of the Study: Aims and Hypotheses

This exploratory study aimed at understanding the contribution of potentially relevant psychosocial determinants of young adults’ pro-environmental behaviors (PEBs) within a comprehensive framework. Besides well-established cognitive factors (i.e., personal values and personal endorsement of a “pro-ecological” worldview implying concern towards environmental problems), we considered the role of affect and identity processes (i.e., feeling of connection to nature, empathy toward nature, and environmental identity). In addition, we analyzed the role of parent–child transmission of ecological values in a variety of young adults’ PEBs, and their underlying mechanisms. According to a bidirectional and dynamic conceptualization of intergenerational value transmission [140,141], we took into account young adults’ ecological values and perceived fathers’ and mothers’ ecological values, that are the values young adults perceive to be important for each of their parents.

Within this framework, we hypothesized a chain relationship between perceived parents’ and mothers’ ecological values, their children’s ecological values, environmental concern, affect towards the environment, and environmental identity, and PEBs, as the final outcomes.

The study was carried out in Italy and involved young adult children. In Italy more and more young adults live with their parents for a long time, thus being widely exposed to parental value socialization. At the same time, young adults are mature and autonomous enough to have responsibilities in the household and the community and the freedom to act as they choose also with respect to pro-environmental behaviors.

## 2. Methods

### 2.1. Materials

An online questionnaire including different scales was developed. A 7-point Likert scale, ranging from 1 = “Completely disagree” to 7 = “Completely agree”, was used to assess the variables. The questionnaire was organized in different sections: the first part provided the informed consent form, contact details of the research team, and general compilation instructions. The sections included validated and well-established scales in the literature for the measurement of the relevant constructs: (1)Twelve items from the environmental portrait value questionnaire (E-PVQ) [148], assessing parental and personal environmental values:(2)Connectedness to nature scale (CNS) [118], measuring the feeling of connection to the natural world;(3)The new environmental paradigm scale (NEP) [72], which was employed as a measure of the personal endorsement of a “pro-environmental” worldview;(4)The dispositional empathy to nature scale (DEN) [120], a measure of empathy towards nature;(5)The environmental identity scale (EID) [109], measuring the relationship between personal identity and nature;(6)The general ecological behavior scale (GEB) [33,34], used to measure PEBs with reference to energy and water conservation, mobility and transportation, waste avoidance, recycling, consumerism, and vicarious behaviors toward conservation.

In the last section, participants were asked for their sociodemographic data. Below is a brief detailed description of the measures employed in the study.

Parental and personal values. The environmental portrait value questionnaire (E-PVQ) [148] was used to assess parental and personal environmental values. The 17-item scale consists of four domains of values, namely biospheric, altruistic, hedonic, and egoistic domains [32,62,67]. For the current study, it was considered only the biospheric domain, which was found to be more significantly associated with PEBs. Respondents were asked to indicate the perceived importance of each value for their parents and themselves (e.g., “It is important to my mother/father to protect the environment”; “It is important to me to protect the environment”).

Connectedness to nature. The connectedness to nature scale (CNS) [118] was employed to estimate to what extent participants feel an emotional connection to the natural world. The 14-item CNS is composed of statements such as “I often feel part of the web of life” and “My personal welfare is independent of the welfare of the natural world”.

New environmental paradigm. The new environmental paradigm scale (NEP) [72] was used to evaluate the personal endorsement of a “pro-environmental” worldview. The 15-item scale provides a measure of the general relationship between humans and nature, and concern for the future of the environment. Example items are “We are approaching the limit of the number of the people the earth can support” and “Mankind was created to rule over the rest of the world”.

Dispositional empathy to nature. The 10-item dispositional empathy to nature scale (DEN) [120] was employed as a measure of the constitutional inclination to comprehend and participate in the natural world’s emotional experience. Four items evaluate perspective-taking (e.g., “I can very easily put myself in the place of the suffering animals and plants”); six items evaluate empathic concern (e.g., “I feel what the suffering animals and plants are feeling”).

Environmental identity. The environmental identity scale (EID) [109] has been used to identify individual differences in the significance of nature for the self-concept. The 24-item scale measures five different aspects of environmental identity: (1) the salience of identity, indicating the degree and significance of an individual’s relationship with nature. (2) The feeling of membership, referring to the mode in which nature has a connection to the community with which one identifies oneself. (3) The acceptance of an ideology shared by the group, reflected by the endorsement of environmental education and a sustainable lifestyle. (4) Positive emotions related to the group, identified by the pleasure felt in nature by means of fulfilment and aesthetic enjoyment. (5) An autobiographical component, comprising memories of interactions with nature. The 11-item short version of the scale proposed by Clayton [109] was used in the present research. Example items are “In general, being part of the natural world is an important part of my self-image” and “When I am upset or stressed, I can feel better by spending some time outdoors ‘communing with nature’”.

General ecological behavior. The 40-item general ecological behavior scale (GEB) [33,34] provides an estimate of general ecological behaviors. Such behaviors are grouped into six domains: energy conservation (e.g., “After one day of use, my sweaters or trousers go into the laundry”), mobility and transportation (e.g., “I ride a bicycle, take public transportation or walk to school”), waste avoidance (e.g., “I buy beverages in cans”), recycling (e.g., “I collect and recycle used paper”), consumerism (e.g., “I eat seasonal produce”), vicarious behaviors toward conservation (e.g., “I have pointed out unecological behavior to someone”). For the current study, some adaptations to the Italian cultural context were made.

### 2.2. Participants and Procedure

Participants were 175 Italian young adults aged 18–30. Detailed socio-demographic data are given in Table 1.

Participants were recruited through a snowball sampling method, and they were asked to answer the online survey on Google form. Data were collected from May to July 2020. The study was conducted in accordance with the ethical standards of the 1964 Declaration of Helsinki, and it fulfilled the ethical standard procedure recommended by the Italian Association of Psychology (AIP). Before taking part in the study, participants were informed of their right to refuse to participate in the study or to withdraw consent to participate at any time during the study without reprisal.

## 3. Results

### 3.1. Preliminary Analyses

*Power analysis*: the a priori power analysis (G*Power 3.1) [149], with alpha = 0.05, power = 0.95 and a medium effect size (ES) of 0.15, showed that the sample size was appropriate for the analysis as the requested sample size was N = 153.

*Normal distribution*: we checked for normal distribution of the study’s variables by analyzing kurtosis and skewness, whose values between −2 and +2 are considered acceptable [150]. The assumption of normality was fully satisfied for most of the variables, except for GEB_recycling, personal values, and CNS. We used the boxplot to identify possible outliers on these variables. Based on the boxplot graphical analysis, we removed one case that was an outlier on all the three variables and two cases that were outliers on two or one of these variables.

The assumption of normality, calculated on the remaining 172 participants, was fully satisfied for all the study’s variables.

### 3.2. Factor and Reliability Analyses

A series of EFAs was conducted in IBM SPSS Statistics (ver. 26) to make an empirical assessment of the dimensionality of the GEB scale [33,34]. Principal component analysis extraction method and the Oblimin oblique rotation with Kaiser Normalization were employed. The KMO measure of sampling adequacy (0.74) and Bartlett’s test of sphericity (approximately, chi-square = 2488.13; df = 780) were both significant (*p* < 0.001). Both the scree plot and factor loadings suggested a four factor solution (Cumulative variance explained = 45.19%; Table 2). The four factors were named: (1) “strong ecological behavior”, indicating pro-environmental behaviors which require a high commitment from the respondent, as in the case of activism (e.g., “I am a member of an environmental organization”); (2) “sustainability in everyday life”, referring to everyday life activities which are performed in a sustainable way (the label was positive while items referred to unsustainable behaviors, e.g., “After one day of use, my sweaters or trousers go into the laundry.”); (3) “recycling and reduced waste production”, including activities oriented to recycling garbage and reducing waste production (e.g., “If possible, I buy products in refillable packages”); (4) “sustainable mobility”, referring to one’s preferential use of ecological or public transportation (e.g., “I usually ride a bicycle, take public transportation or walk to go to the university/at work”). The dimensions were not significantly correlated. Items showing a factor loading >0.40 were retained. The following items did not show a significant factor loading onto any factor, and were thus excluded from the final solution: I leave electrically powered appliances (TV, stereo, printer) on standby; In the winter, if possible, I turn down the heat when I leave my room for several hours; I buy beverages in cans; If possible, I buy beverages in returnable bottles; I refrain from battery-operated appliances; I keep gift wrapping paper for reuse; If possible, I eat seasonal products; I use writing pads from recycled paper; If my parents should ever change the car, I would try to persuade them to buy an energy-efficient one; I insist on holidays close to home.

Reliability analyses of the scales’ items were conducted using Cronbach’s alpha coefficient. The internal consistency showed an adequate level of reliability for all the scales [151]. Cronbach’s alpha was found to be good for each of the four dimensions emerging from EFAs on the GEB scale (α > 0.70); very good for E-PVQ, CNS, EID, and DEN, ranging from 0.88 to 0.95; and good for the NEP (α = 0.72).

### 3.3. Correlational Analysis (CA) and Hierarchical Multiple Regression Analysis (HMRA)

The correlation matrix showed several significant correlations between the variables considered (Table 3). Among the different dimensions, paternal values were found to be highly correlated with both maternal (r = 0.724; *p* < 0.01) and personal values (r = 0.410; *p* < 0.01). Personal values showed a significant association with CNS (r = 0.694; *p* < 0.01), EID (r = 0.678; *p* < 0.01) and GEB (r = 0.510; *p* < 0.01); CNS was also correlated with paternal (r = 0.287; *p* < 0.01) and maternal values (r = 0.357; *p* < 0.01); DEN showed the highest correlation with CNS (r = 0.547; *p* < 0.01) and personal values (r = 0.398; *p* < 0.01), whereas EID emerged to have a significant association with CNS (r = 0.749; *p* < 0.01), personal values (r = 0.678; *p* < 0.01) and DEN (r = 0.570; *p* < 0.01); GEB was strongly correlated with CNS (r = 0.565; *p* < 0.01), EID (r = 0.516; *p* < 0.01), and personal values (r = 0.510; *p* < 0.01). NEP was not found to be significantly associated with any of the variables considered, with the exception of DEN (r = 0.342; *p* < 0.01), GEB (r = 0.202; *p* < 0.01), and CNS (r = 0.173; *p* < 0.05).

Finally, hierarchical multiple regression analyses (HMRAs) were performed in order to examine the association between the predictor variables and the four PEBs emerging from factor analyses: strong ecological behavior (Table 4), sustainability in everyday life (Table 5), recycling and reduced waste production (Table 6), sustainable mobility (Table 7).

In order to identify the added value of parental values, personal values and worldviews, emotional variables, and identity, in a chain relationship according to our hypotheses, the independent variables have been included in the model in the following order: paternal and maternal values were included in step 1, personal values and NEP in step 2, CNS and DEN in step 3, and EID in step 4.

*HMRA on strong ecological behaviors* (Table 4). At step 1, the model was significant and paternal values appeared to be the only predictor of strong ecological behaviors. At step 2, the model was significant, showing a considerable increase in the amount of explained variance; paternal and personal values emerged as significant predictors of strong ecological behaviors. At step 3, the model still increased the amount of explained variance; DEN and CNS appeared to be significant predictors of strong ecological behaviors; paternal values were still significant whereas personal values were no longer a significant predictor. Finally, at step 4, the model still showed an increasing amount of explained variance, with only paternal values and EID as final predictors.

*HMRA on sustainability in everyday life* (Table 5). At step 1, the model was not significant, and none of the variables considered emerged as predictors of sustainability in everyday life. At step 2, The model was significant. Personal values appeared to be the only significant predictor of sustainability in everyday life. At step 3, the model still showed an increase in explained variance, with CNS emerging as a significant predictor of sustainability in everyday life, while personal values were no longer significant. Finally, at step 4, no further significant change emerged in the model, and CNS was still found as the only predictor of Sustainability in Everyday Life.

HMRA on Recycling and reduced waste production (Table 6). At step 1, the model was significant and maternal values emerged as the only predictor of recycling and reduced waste production. At step 2, the model showed a considerable increase in the amount of explained variance, with personal values and NEP emerging as significant predictors of recycling and reduced waste production, and maternal values no longer significant. At both step 3 and 4, the model did not show further increase in explained variance. Personal values and NEP still emerged as significant predictors of recycling and reduced waste production in the final model.

HMRA on sustainable mobility (Table 7). At step 1, the model was not significant. At step 2, the model showed a tendency to significance (*p* < 0.10), with personal values emerging as a predictor of sustainable mobility. At both steps 3 and 4, the model did not show any further increase in explained variance and no significant predictor of sustainable mobility was retained in the final model.

## 4. Discussion

This study aimed at gaining a better understanding of the psychosocial determinants of young adults’ pro-environmental behaviors (PEBs), adding the contribution of affect and identity processes to well-established cognitive factors, and the role of intergenerational transmission of ecological values in these mechanisms.

Past research has identified a variety of cognitive factors that can affect PEBs. The role of personal values has been studied worldwide since decades. Self-transcendence and altruistic values have been repeatedly found to predict a variety of PEBS [31,56,58,152]. More recently, de Groot and Steg [63] argued how values specifically referring to the conservation of the environment, namely biospheric values, can be more enduringly associated with PEBs. Consistent empirical evidence has also supported the importance of environmental concern, often measured through the NEP scale [71,72], in different PEBs [74,77,153]. Finally, the role of different norms in PEBs, either as distinct variables or included in models such as NAT [80], VBN [32], and TPB [86], has often emerged in the literature [59,63,81,82,83,84,88].

With reference to affect, different studies have outlined that specific PEBs, such as reduction of car use, or use of electric cars, may be also based on affective motives [98,99,100,101]. More recently, the role of a more general emotional relationship with nature in PEBs has been suggested [121,122,123,124,125]. Identity processes have been included in the realm of variables predicting PEBs as well. Traditionally, the role of identity in PEBs has been considered within the Social Identity Theory (SIT) [154,155,156], according to which individuals may conform to the values, beliefs, and behaviors of the social groups they belong to. Within this framework, identity has emerged to improve the predictive power of well-established models referring to PEBs, including TPB [102, 105,108,113]. More specifically, some authors have stressed that beyond belonging to social groups whose norms are related to PEBs, it is the general inclusion of the environment in the self, as conceptualized in environmental identity [109], which can be more strongly predictive of PEBs. Several studies have supported this claim, with reference to a variety of PEBs [103,110,111,112,113,114,118].

The present study has combined all the above determinants in a more comprehensive framework, with reference to different PEBs, with interesting results. First of all, the measure of GEB scale [33,34] has yielded four reliable dimensions from the analysis of our respondents, partially overlapping with the original distinction. Vicarious behaviors toward conservation, energy and water conservation, waste avoidance, recycling, consumerism, mobility and transportation, turned into strong ecological behaviors, indicating pro-environmental behaviors requiring strong levels of commitment, and implying active participation in environmental organizations and pressure on other people towards more sustainable behaviors; sustainability in everyday life, referring to everyday life activities which have to do with energy conservation; recycling and reduced waste production, including activities oriented to garbage recycling and the reduction of waste production; and sustainable mobility, referring to the use of ecological or public transportation. This distinction embraces domains of PEBs largely identified in previous research [29,38].

With reference to each of those PEBs, hierarchical multiple regression analyses have identified specific patterns of association with cognitive, affective, and identity predictors. Strong ecological behaviors were directly predicted by paternal values, while the role of personal values was significant at the first step of the analysis and became not significant when the affective variables referring to connectedness to nature and dispositional empathy to nature, and environmental identity, were included in the model. This pattern is consistent with previous findings on the role of values and identity processes in environmental activism [32,66,108,157]. Interestingly, the role of emotions in environmental activism has emerged in a handful of studies [158,159,160]. Sustainability in everyday life was predicted by personal values, whose role became not significant when connectedness with nature was included in the model. This result is in line with the increasing interest in the role of emotional bonds with the environment in PEBs. A recent meta-analysis on 37 samples from 26 studies has shown that a feeling of connection with nature is positively associated with PEBs, while the geographic location of a study, and the age and gender of participants were not significant moderators [161]. Recycling and reduced waste production were predicted by maternal values, whose role became not significant when personal values and environmental concern measured through the NEP scale were included in the model. Sustainable mobility was found to be predicted by personal values, which became not significant in the following steps of the analysis. Personal values and environmental concern have consistently emerged as determinants of PEBs related to recycling and waste reduction in the literature [162,163]. TPB, in both basic and extended versions, has been fruitfully applied worldwide to predict sustainable mobility [164,165,166]. With reference to these PEBs, the emotional connection with nature and environmental identity did not emerge as significant predictors. This outcome may be related to the widespread knowledge about these environmental problems since decades [167]. In this regard, cognitive variables may play a much more relevant role. Further research is undoubtedly needed to better understand this issue. Overall, a relevant role of the emotional connection with nature has emerged across different behaviors, suggesting the importance of integrating non-rationale variables to previous models predicting PEBs [121,122,125]. Moreover, these results seem to suggest a mediation role of some of these variables, which should be tested in future research with larger samples and is compatible with recent research findings [123]. Moreover, environmental identity has emerged to be a relevant predictor, but with differences between PEBs, and with reference to strong ecological behaviors above all, as discussed above. The general result on a different role of identity processes in performing PEBs is in line with previous findings [103,113]. Future research will have to clarify whether identity does not play a role in some PEBs at all, or whether different mechanisms referring to identity should be considered.

One of the most innovative processes considered in the present study was the intergenerational transmission of ecological values, here analyzed in terms of perceived fathers’ and mothers’ biospheric values. From the regression results it has emerged that perceived parental values were significantly related to some of the young adults’ PEBs. In particular, perceiving that biospheric values were important to fathers was the most relevant predictor of young adults’ strong ecological behaviors. Differently, maternal environmental values were more relevant in predicting young adults’ recycling and reduced waste production. The more that young adults perceived that their mothers attributed importance toward protecting environment and natural resources, the more likely they were to recycle and reduce waste production. These findings confirmed previous studies about the differential impact of fathers and mothers on their children’s behaviors [143,168]. Fathers, who are generally more likely to encourage children’s instrumental behaviors [146], seem able to promote children’s strongly goal-oriented actions to reach more socially sustainable behaviors. Mothers’ biospheric values, instead, contribute to children’s recycling activities and waste reduction, by suggesting that mothers act as role models especially for daily pro-environmental behavior within the family context. Indeed, as shown by previous research [146], mothers, more than fathers, tend to encourage children’s involvement in domestic life.

Both in the case of strong ecological behavior and recycling and reduced waste production, young adults’ personal biospheric values were positively linked to the outcome. Together with perceived parental values, children’s internalized values give a substantial contribution to pro-environmental behaviors. As already mentioned above, it is not surprising that young adults, involved in achieving autonomy and establishing identity, are likely to act consistently with their own values. More interestingly, from a systemic perspective, we could speculate that young adults who feel to share with their fathers and mothers the importance of biospheric values are more prone to perform sustainable behaviors within and outside the family.

### Limitations and Future Research

Four main limitations of the present study must be acknowledged. First, the study’s cross-sectional design did not allow us to draw causal interpretations from the results or to catch potential changes over time. Second, the sample was too small to test more complex models. It was of convenience and sampled through a snowball method. This sampling method limits the possibility to generalize the study findings to the whole population. It is likely that the more motivated youth or youth interested in environmental issues decided to take part in the study. To overcome this limitation, during the data collection, we proposed to reach a certain degree of sample diversity by beginning the sample within data collecting contexts that were as diverse as possible. Moreover, we tried to motivate reluctant respondents to participate in the study. Third, no data about parents’ actual biospheric values were collected, but only the young adults’ perceptions of their parents’ values were analyzed. Although perceived parental values are an important step in the value transmission process [141], the literature on interpersonal perceptions in parent–child relationship highlights the risk of the so-called consensus bias [169], which is the tendency of children (or parents) to believe that their values, attitudes, affect, etc., are shared by the other family members. To account for this projection effect, future studies should control for parents’ actual values. Furthermore, the role of other relevant psychosocial variables associated with PEBs and identified by the literature should be considered, with personal norms standing first. In our study, we considered personal values and environmental identity, which has been conceived as a proximal variable predicting PEBs [32,110], but not personal norms. Our results seem to suggest that personal norms and environmental identity may play a role on a distinct, although interactive, level. A better understanding of the relationships between them should be the focus of future research. Finally, additional research should be aimed at better understanding the intergenerational transmission of biospheric values: further investigations might explore the role of family dynamics in determining the quality of the processes mentioned above.

## 5. Conclusions

This study developed an analysis of cognitions, emotions, and identity processes associated with performing pro-environmental behaviors in young adults. From a theoretical standpoint, it provides a more integrative view of pro-environmentally related behaviors by considering the large variety and different nature of human processes. Moreover, it stresses the importance to pay greater attention to family dynamics, by adopting an intergenerational perspective, to better understand and support young generations’ PEBs. It also has practical implications in terms of family relationships and education processes. On the one hand, parents should promote communication about the importance of environmental issues and act as reliable behavioral models in order to raise biospheric values in children; on the other, the importance of nurturing emotions towards nature and including the environment in the definition of the self in the education of the youngest generations should be recognized in educational systems. At school, the development of ecological values, attitudes, and feelings is facilitated through activities that introduce alternative teaching and learning techniques, quite far from the frontal teaching of the classical school [170]. In this regard, the importance of direct contact with nature in promoting PEBs through these mechanisms has been recently suggested [171,172]. Rosa et al. [172], for example, showed that greater contact with nature during childhood is associated with greater contact with nature as an adult, which, in turn, promotes connectedness to nature and self-reported PEB. What the results of our study suggest is to move towards a more holistic approach in school education by eliciting cognitive, emotional, and identity processes to support different PEBs (from strong ecological behavior to daily pro-environmental behaviors) within a value alliance with the family.

## Figures and Tables

**Table 1 ijerph-19-01670-t001:** Socio-demographic variables of participants.

Socio-Demographic Variables	%	Mean	St. Dev.
**Sex**			
Men	43.6		
Women	56.4		
**Age**		24.84	2.56
**Education**			
High school	36.6		
Bachelor’s degree	38.4		
Master’s degree	20.9		
Ph.D.	4.1		
**Place of residence**			
Northern Italy	16.5		
Central Italy	59.2		
Southern Italy	24.3		
**Work activity**			
Student	68.8		
Office worker	10.5		
Teacher	4.7		
Unemployed	3		
Other	13		

**Table 2 ijerph-19-01670-t002:** EFA: results and factors loadings for GEB scale.

Item	Factor 1	Factor 2	Factor 3	Factor 4
	Strong Ecological Behavior	Sustainability in Everyday Life	Recycling and Reduced Waste Production	Sustainable Mobility
I contribute financially to environmental organizations	0.738			
I am a member of an environmental organization	0.674			
I read books, publications, and other materials about environmental problems	0.611			
When shopping, I prefer products with eco-labels	0.550			
If possible, I buy certified organic foods	0.533			
I learn about environmental issues in the media (newspapers, magazines, and TV)	0.483			
I have pointed out unecological behavior to someone	0.461			
I ask my parents to buy seasonal produce	0.453			
On excursions, I take along beverages in single-use packages		−0.763		
I normally use plastic bags		−0.683		
At my parties, we use plastic silverware and paper cups		−0.676		
After one day of use, my sweaters or trousers go into the laundry		−0.585		
I put empty batteries in the garbage		−0.556		
In hotels, if possible, I have the towels changed daily		−0.533		
In the winter, it is warm enough in my room to only wear a T-shirt		−0.508		
I eat in fast-food restaurants, such as McDonalds and Burger King		−0.495		
I order take-out pizza		−0.456		
At home, I kill insects with a chemical insecticide		−0.454		
I collect and recycle used paper			0.727	
I separate waste			0.708	
I reuse my shopping bags			0.680	
I bring empty glass bottles to a recycling bin			0.664	
As the last person to leave a room, I switch off the lights			0.652	
After a picnic, I leave the place as clean as it was before			0.574	
I buy products in refillable packages (ex.: once used up the whole product content, I refill the old flacon without buying a new one)			0.433	
For making notes, I take paper that is already used on one side			0.432	
When I need to get around the city, I usually use public transport				0.782
I usually ride a bicycle, take public transportation or walk to go to the university/at work				0.777
To cover short distances, I usually walk or ride a bicycle				0.672
When possible, I refrain from using the car				0.611
Eigenvalues	5.74	2.99	2.58	2.25
Explained variance	19.1	29.09	37.7	45.19
Mean	3.19 ± 1.03	3.85 ± 1.05	5.28 ± 0.76	4.25 ± 1.46
Cronbach’s Alpha	0.78	0.75	0.79	0.77

**Table 3 ijerph-19-01670-t003:** Correlation analysis: ecological values, NEP, CNS, DEN, EID, and GEB.

	1	2	3	4	5	6	7	8
1. Paternal values	1							
2. Maternal values	0.724 **	1						
3. Personal values	0.410 **	0.528 **	1					
4. NEP	0.092	0.066	0.091	1				
5. CNS	0.287 **	0.357 **	0.694 **	0.173 *	1			
6. DEN	0.173 *	0.158 *	0.398 **	0.342 **	0.547 **	1		
7. EID	0.296 **	0.368 **	0.678 **	0.096	0.749 **	0.570 **	1	
8. GEB	0.227 **	0.242 **	0.510 **	0.202 **	0.565 **	0.421 **	0.516 **	1

*: *p* < 0.05; **: *p* < 0.01.

**Table 4 ijerph-19-01670-t004:** HMRA: predictors of strong ecological behavior.

Predictors	β Coefficients				Adjusted R^2^	R^2^ Change
	Step 1	Step 2	Step 3	Step 4		
**Step 1**					0.097 ***	
Paternal values	0.288 **					
Maternal values	0.052					
**Step 2**					0.240 ***	0.146 ***
Paternal values		0.256 **				
Maternal values		−0.164				
Personal values		0.445 ***				
NEP		0.070				
**Step 3**					0.334 ***	0.105 ***
Paternal values			0.237 **			
Maternal values			−0.128			
Personal values			0.189			
NEP			−0.027			
CNS			0.232 *			
DEN			0.233 **			
**Step 4**					0.370 ***	0.038 **
Paternal values				0.238 **		
Maternal values				−0.143		
Personal values				0.099		
NEP				0.004		
CNS				0.090		
DEN				0.148		
EID				0.334 **		

***: *p* < 0.001; **: *p* < 0.01; *: *p* < 0.05.

**Table 5 ijerph-19-01670-t005:** HMRA: predictors of sustainability in everyday life.

Predictors	β Coefficients				Adjusted R^2^	R^2^ Change
	Step 1	Step 2	Step 3	Step 4		
**Step 1**					−0.005	
Paternal values	−0.035					
Maternal values	0.103					
**Step 2**					0.059 **	0.074 **
Paternal values		−0.062				
Maternal values		−0.038				
Personal values		0.291 ***				
NEP		0.101				
**Step 3**					0.099 ***	0.050 *
Paternal values			−0.061			
Maternal values			−0.039			
Personal values			0.085			
NEP			0.085			
CNS			0.337 **			
DEN			−0.068			
**Step 4**					0.094 ***	0.000
Paternal values				−0.060		
Maternal values				−0.040		
Personal values				0.080		
NEP				0.087		
CNS				0.328 **		
DEN				−0.074		
EID				0.021		

***: *p* < 0.001; **: *p* < 0.01; *: *p* < 0.05.

**Table 6 ijerph-19-01670-t006:** HMRA: predictors of recycling and reduced waste production.

Predictors	β Coefficients				Adjusted R^2^	R^2^ Change
	Step 1	Step 2	Step 3	Step 4		
**Step 1**					0.082 ***	
Paternal values	0.033					
Maternal values	0.280 **					
**Step 2**					0.269 ***	0.193 ***
Paternal values		−0.013				
Maternal values		0.069				
Personal values		0.436 ***				
NEP		0.216 ***				
**Step 3**					0.276 ***	0.015
Paternal values			−0.019			
Maternal values			0.081			
Personal values			0.325 ***			
NEP			0.181 **			
CNS			0.119			
DEN			0.071			
**Step 4**					0.274 ***	0.003
Paternal values				−0.020		
Maternal values				0.085		
Personal values				0.349 ***		
NEP				0.173 *		
CNS				0.155		
DEN				0.093		
EID				−0.086		

***: *p* < 0.001; **: *p* < 0.01; *: *p* < 0.05.

**Table 7 ijerph-19-01670-t007:** HMRA: predictors of sustainable mobility.

Predictors	β Coefficients				Adjusted R^2^	R^2^ Change
	Step 1	Step 2	Step 3	Step 4		
**Step 1**					−0.011	
Paternal values	0.005					
Maternal values	0.014					
**Step 2**					0.005	0.028 °
Maternal values		−0.009				
Maternal values		−0.078				
Personal values		0.190 *				
NEP		0.033				
**Step 3**					0.016	0.022
Paternal values			−0.019			
Maternal values			−0.058			
Personal values			0.079			
NEP			−0.018			
CNS			0.080			
DEN			0.136			
**Step 4**					0.010	0.000
Paternal values				−0.019		
Maternal values				−0.058		
Personal values				0.082		
NEP				−0.019		
CNS				0.085		
DEN				0.139		
EID				−0.012		

*: *p* < 0.05; °: *p* < 0.10.

## Data Availability

The data presented in this study are available upon request from the corresponding author.

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
