# Peer review of "My Parents Taught…Green Was My Growth! The Role of Intergenerational Transmission of Ecological Values in Young Adults’ Pro-Environmental Behaviors and Their Psychosocial Mechanisms"

_ijerph, 2022, doi:10.3390/ijerph19031670_

Round 1

Reviewer 1 Report

The article is focused on interesting and worth-investigating relations between pro-environmental behaviors (PEBs) and their determinants, including how parents’ ecological values can affect PEBs of their children. The manuscript is generally well written, structured and referenced. The methodology is presented in a proper way. However, for an article to be published, the authors should address the following issues.

  1. The authors referred to the limitations of the study (lines 605-621), presenting four of them: 1/ cross-sectional design of the study, 2/ the convenience of the sample, 3/ lack of data about parents’ actual biospheric values and 4/ lack of the analysis of the role of some relevant psychological variables associated with PEBs identified in the literature. In my opinion the second limitation is presented only in a general way. I strongly suggest the authors to refer to the limitations of snowball method of sampling and to explain how they tried (what they did, if any) to overcome this limitation in their research process. It is also necessary to explain how (in what way) the snowball method of sampling could affect the results of the study.

  1. I would recommend the authors to extend the Conclusions section by explaining more precisely what are practical implications of the study. I mean explaining, for example, in what way the results of the study may be used by educational institutions.

Author Response

Dear Reviewer,

First, we would like to thank your thoughtful and constructive comments, which led to a considerable improvement of the manuscript. We trust that all of your remarks have been addressed with appropriate changes in the revised manuscript.

In the following, we provide you with a point-by-point reply to your comments:

Comment 1. The authors referred to the limitations of the study (lines 605-621), presenting four of them: 1/ cross-sectional design of the study, 2/ the convenience of the sample, 3/ lack of data about parents’ actual biospheric values and 4/ lack of the analysis of the role of some relevant psychological variables associated with PEBs identified in the literature. In my opinion the second limitation is presented only in a general way. I strongly suggest the authors to refer to the limitations of snowball method of sampling and to explain how they tried (what they did, if any) to overcome this limitation in their research process. It is also necessary to explain how (in what way) the snowball method of sampling could affect the results of the study.

Authors’ response. As suggested, in the revised version of the manuscript we have referred to the limitations of the snowball method of sampling and to what we did to overcome its main limitations:

“Second, the sample was too small to test more complex models. It was of convenience and sampled through a snowball method. This sampling method limits the possibility to generalize the study findings to the whole population. It is likely that the more motivated youth or youth interested in environmental issues decided to take part in the study. To overcome this limitation, during the data collection we proposed to reach a certain degree of sample diversity by beginning the sample within data collecting contexts that were as diverse as possible. Moreover, we tried to motivate reluctant respondents to participate in the study (lines 643-650).

Comment 2. I would recommend the authors to extend the Conclusions section by explaining more precisely what are practical implications of the study. I mean explaining, for example, in what way the results of the study may be used by educational institutions.

Authors’ response. In the revision of the manuscript, we have extended the practical implications of the study findings:

“At school, the development of ecological values, attitudes, and feelings is facilitated through activities that introduce alternative teaching and learning techniques, quite far from the frontal teaching of the classical school [170]. In this regard, the importance of direct contact with nature in promoting PEBs through these mechanisms has been recently suggested [171,172]. Rosa et al. [172], for example, showed that greater contact with nature during childhood is associated with greater contact with nature as an adult, which, in turn, promotes connectedness to nature and self-reported PEB. What the results of our study suggest is to move towards a more holistic approach in school education by eliciting cognitive, emotional, and identity processes to support different PEBs (from strong ecological behavior to daily pro-environmental behaviors) within a value alliance with the family” (682-692).

Reviewer 2 Report

The structure of the reviewed article is correct. The authors described in detail what the topic of the article was, presented the research results and held a discussion.
Critical remarks on the article:
- the abstract should contain a clearly defined research goal, the adopted research methodology and the main conclusions of the research. In my opinion, it requires a significant correction;
- there is no indication of the limitations of research;
- no definition of further research directions;
- the quoted literature should cover the period after 2000.

Author Response

Dear Reviewer,

First, we would like to thank your thoughtful and constructive comments, which led to a considerable improvement of the manuscript. We trust that all of your remarks have been addressed with appropriate changes in the revised manuscript.

In the following, we provide you with a point-by-point reply to your comments:

  1. The structure of the reviewed article is correct. The authors described in detail what the topic of the article was, presented the research results and held a discussion.

Reply: Thank you. We highly appreciated your positive comment.

  1. The abstract should contain a clearly defined research goal, the adopted research methodology and the main conclusions of the research. In my opinion, it requires a significant correction;

Reply:  We agree that the previous version of the abstract lacked several important information. Thus, we have revised the abstract according to your suggestions, as follows (please find the main points of your previous comment in italics within brackets): 

Past research on pro-environmental behaviors (PEBs) has identified several psychosocial determinants, from personal values to attitudes – mostly environmental concern – and norms. Less attention has been devoted to the role of affect and identity processes until recently, when the emotional connection with nature and environmental identity -  i.e. one's self-concept in relation to the natural world - have started being investigated. Finally, an interesting research line on parent-child transmission of ecological values has been recently developed. We aimed to analyze the role of the above-mentioned variables in predicting different PEBs, within a comprehensive framework. We hypothesized a chain relationship between parents’ and mothers’ ecological values, their children’s ecological values, environmental concern, affect towards the environment, and environmental identity, and PEBs, as the final outcome.  [Research goal and hypotheses]. In a cross-sectional exploratory study [Research Methodology], an online questionnaire was administered to 175 Italian young adults. Validated scales to measure the above variables and socio-demographics were included. Results showed different patterns of predictors for each PEB. Overall, the importance of the emotional connection with nature and environmental identity in predicting PEBs has clearly emerged. Finally, the role of intergenerational transmission of environmental values in PEBs, with differences between the influence of fathers and mothers, has been outlined. The study provides a more integrative view of PEBs by considering the variety of human processes [Main conclusions of the research]. Theoretical and practical implications of results are discussed.

  1. There is no indication of the limitations of research and 4. No definition of further research directions

Reply:  As a matter of fact, we have discussed both limitations and directions for future research in section 4. Discussions. Nevertheless, we totally agree with your remark in that the part about the limitations could have been more detailed. Moreover, we realized that it would have been more accurate to describe them in a separate section. In order to improve the readability of the manuscript, a new subsection named 4.1. Limitations and future research has been added (lines 640-667). We also have provided further information about the limitations of the employment of a snowball method of sampling (lines 643-650), as suggested also by Reviewer 1. Finally, we added a new research issue in addition to those already mentioned (lines 664-667). We thank you for this highly appreciated suggestion.

  1. The quoted literature should cover the period after 2000

Reply:  We concur with you on the importance of citing very recent literature. We are well aware that our field of research is evolving very fast. For those reasons, the large majority of the articles we cited cover the period 2000-2021. However, we also wished to include some works that went on to become milestones for our discipline, most of which were produced before the 2000s’. However, they represent about 17% of the total amount of references.

Reviewer 3 Report

On materials, a clear statistical statement (add demographic table) is required for the survey.

In the measures, please provide more specific reasons for selecting the scale in relation to this research hypothesis.

Please specify the research model and hypothesis. In addition, please write the contents of the hypothesis background in the theoretical background separately from theoretical background

Confirm this Chi-Square = 2488,13 -> 2488.13

Please mark questions below 0.5 deleted from EFA.

Please review the statistical results on Hierarchical Multiple Regression Analysis 

Please organize the analysis result table according to the hypothesis.

Author Response

Dear Reviewer,

First, we would like to thank your thoughtful and constructive comments, which led to a considerable improvement of the manuscript. We trust that all of your remarks have been addressed with appropriate changes in the revised manuscript.

In the following, we provide you with a point-by-point reply to your comments:

R3: On materials, a clear statistical statement (add demographic table) is required for the survey.

A table with sociodemographic variables was included in the manuscript in the Section "Participants and Procedure".

In the measures, please provide more specific reasons for selecting the scale in relation to this research hypothesis.

We added a sentence explaining that the tools we adopted for this study are validated and well-established scales in the literature (lines 329-330).

Please specify the research model and hypothesis. In addition, please write the contents of the hypothesis background in the theoretical background separately from theoretical background

The final section of the Introduction has been now entitled "Overview of the study: Aims and hypotheses". Hypotheses were now better indicated in this section (lines 313-316.

Confirm this Chi-Square = 2488,13 -> 2488.13

Right! Done.

Please mark questions below 0.5 deleted from EFA.

We added this information in the text at the end of the Section Results – Factor and Reliability Analyses (lines 450-456)

Please review the statistical results on Hierarchical Multiple Regression Analysis

We have reviewed the Results Section referring to HMRAs to match the content of the text and the tables.

Please organize the analysis result table according to the hypothesis.

We have better expressed in the Section "Overview of the study: aims and Hypotheses" that we expect a chain relationship between the variables, in which each step of the analysis adds in terms of explained variance. The presentation of the Tables referring to HMRAs is in line with this organization.